# Sarcoid Nodule or Lung Cancer? A High-Resolution Computed Tomography-Based Retrospective Study of Pulmonary Nodules in Patients with Sarcoidosis

**DOI:** 10.3390/diagnostics14212389

**Published:** 2024-10-26

**Authors:** Chiara Catelli, Susanna Guerrini, Miriana D’Alessandro, Paolo Cameli, Antonio Fabiano, Giorgio Torrigiani, Cristiana Bellan, Maria Antonietta Mazzei, Piero Paladini, Luca Luzzi

**Affiliations:** 1Lung Transplant Unit, Department of Medical, Surgical and Neuro-Sciences, Azienda Ospedaliero-Universitaria Senese, University of Siena, 53100 Siena, Italy; luca.luzzi@unisi.it; 2Diagnostic Imaging Unit, Department of Medical, Surgical and Neuro-Sciences, Azienda Ospedaliero-Universitaria Senese, University of Siena, 53100 Siena, Italy; guerrinisus@gmail.com (S.G.); giorgio.torrigiani@student.unisi.it (G.T.); mariaantonietta.mazzei@unisi.it (M.A.M.); 3Respiratory Disease and Lung Transplant Unit, Department of Medical, Surgical and Neuro-Sciences, Siena University, 53100 Siena, Italy; dalessandromiriana@gmail.com (M.D.); paolo.cameli@unisi.it (P.C.); 4Thoracic Surgery Unit, Department of Medical, Surgical and Neuro-Sciences, Azienda Ospedaliero-Universitaria Senese, University of Siena, 53100 Siena, Italy; fabianoantonio092@gmail.com (A.F.); chirtoracicasiena@gmail.com (P.P.); 5Institute of Pathological Anatomy and Histology, Department of Medical Biotechnologies, Azienda Ospedaliero-Universitaria Senese, University of Siena, 53100 Siena, Italy; cristiana.bellan@unisi.it

**Keywords:** lung nodule, pulmonary sarcoidosis, lung cancer, lung adenocarcinoma, computed tomography, lymph nodes, HRCT, lung resection

## Abstract

**Background**: The objective of this retrospective study was to compare the characteristics of sarcoid nodules and neoplastic nodules using high-resolution computed tomography (HRCT) in sarcoidosis patients. **Methods**: This is a single-center retrospective study. From 2010 to 2023, among 685 patients affected by pulmonary sarcoidosis, 23 patients developed pulmonary nodules of a suspicious malignant nature. The HRCT characteristics of biopsy-proven malignant (Group A) vs. inflammatory (Group B) nodules were analyzed and compared. **Results**: A significant difference was observed between the groups in terms of age (*p* = 0.012). With regard to HRCT features, statistical distinctions were observed in the appearance of the nodule, more frequently spiculated in the case of lung cancer (*p* < 0.01), in the diameter of the nodule (Group A: 23.5 mm; Group B: 12.18 mm, *p* < 0.02), in the median nodule density (Group A: 60.0 HU, Group B: −126.7 HU, *p* < 0.01), and in the number of pulmonary nodules, as a single parenchymal nodule was more frequently observed in the neoplastic patient group (*p* = 0.043). In Group A, the 18-PET-CT demonstrated hilar/mediastinal lymphadenopathy in 100% of cases; histology following surgery did not report any cases of malignant lymph node involvement. **Conclusions**: An accurate clinical evaluation and HRCT investigation are crucial for diagnosing lung cancer in patients with sarcoidosis in order to determine who requires surgical resection. The spiculated morphology of the nodule, greater size, the number of pulmonary nodules, and density using HRCT appear to correlate with the malignant nature of the lesion.

## 1. Introduction

Sarcoidosis is a systemic inflammatory disease characterized by the formation of noncaseating epithelioid cell granulomas in multiple organs. X-ray is the first diagnostic study performed, followed by high-resolution computed tomography (HRCT) [1]. Typical morphological patterns comprise bilateral nodules with distribution in the peribronchovascular interstitium, costal pleura, and interlobar fissures, accompanied by bilateral and symmetric hilar lymphadenopathy. Other common findings include bilateral consolidations or ground glass opacities. Atypical radiological presentations may occur in 15–25% of sarcoidosis cases (2), including lotus–torus-like sarcoidosis, unilateral lymphadenopathy, fibrosis [2], a solitary lung mass [3], or a parenchymal nodular variant [4]. The latter has an incidence of 4%, more prevalent in women [5]. A definitive diagnostic test for sarcoidosis does not yet exist, and the diagnosis is made by exclusion, based on compatible clinical and radiological manifestations. Occasionally, histopathological characterization from the lymph node or lung tissue is possible through biopsies from bronchoscopy, transbronchial needle aspiration (TBNA), or surgery. Differentiating between lung cancer (LC) and sarcoidosis can be challenging, especially as the association between these two conditions is becoming more apparent [6]. The 18-FDG-PET scan is not a reliable test, as the inflammatory nature of sarcoid lesions can show uptake similar to LC [7]. Perfusion computed tomography (CTp) emerges as a viable technique for reproducible measurements in solid nodules (>10 mm). CTp has the potential to decrease the indeterminate diagnoses of solid nodules mitigating the need for follow-up CT scans, 18-FDG-PET studies, biopsy, or unwarranted surgical interventions [8]. However, it does not represent a widely used and universally accepted technique. Currently, there is still insufficient literature to support the risk of LC in patients with sarcoidosis; although, an increased relative risk of LC within 4–6 years after the diagnosis of sarcoidosis has been demonstrated [9], possibly induced by chronic inflammation in the affected lungs. Moreover, LC patients can develop a sarcoid-like reaction due to an immune response related to cancer antigens or other factors. In some cases, sarcoidosis or LC can arise independently and show similar clinical and radiological characteristics [10]. The purpose of our study is to compare the radiological characteristics of suspicious parenchymal nodules in patients diagnosed with sarcoidosis, in order to identify which of these may indicate the more probable heteroplastic nature of the nodule. A secondary goal is to outline an appropriate diagnostic and therapeutic approach, in order to only refer for surgery those patients in whom differential diagnosis has not been possible through other means and, especially, for patients for whom the neoplastic suspicion justifies a high-risk surgical procedure.

## 2. Materials and Methods

Between 1 January 2010 and 1 January 2023, 685 patients with pulmonary sarcoidosis were included within follow-up at our “Rare Respiratory Diseases” referral center. Among these patients, 72 had pulmonary involvement of the nodular variant of sarcoidosis, with at least one nodule of a suspicious neoplastic nature. During the **radiologic 3-months** follow-up, 49 patients had shown a reduction in the number and size of lung nodules following changes in medical therapy, demonstrating the benign nature of those lesions, and were, thus, excluded from this study. The remaining 23 patients with pulmonary nodules suspected to be neoplastic and with CT examinations of good technical quality were enrolled in the study after histopathological characterization. 

The sarcoid pathology stage, according to Scadding classification [11], was determined based on a chest X-ray executed in the previous 4 months before nodule identification on a CT scan.

All patients underwent histopathological confirmation through CT-guided fine-needle aspiration biopsy (FNAB), a video-assisted thoracoscopy (VATS) surgical biopsy, open thoracotomy surgical biopsy, or robotic-assisted thoracoscopy (RATS) biopsy.

The present study was approved by the hospital’s Institutional Review Board, and patients were requested to provide explicit consent by signing a specific agreement, permitting the retrospective examination of their medical records for scientific research purposes.

### 2.1. Thoracic CT Scan

The radiological follow-up at our center consisted of annual HRCT scans for stable lung sarcoidosis or more frequent monitoring (every 6–8 months) in cases of the exacerbation or progression of sarcoidosis or alterations in the usual therapeutic approach. In case of new-onset suspicious nodules a 3-months follow-up was adopted. All the CT scans were obtained with the patient in the supine position with a 64-detector row configuration (Discovery CT 750HD, General Electric Healthcare, Wauwatosa, WI, USA and VCT, General Electric Healthcare, Wauwatosa, WI, USA) in the helical mode, from the base of the lung to the thoracic inlet. In 16/23 patients, the CT examination was conducted without contrast medium; in the 7/23 patients with scans, the examination was carried out with and without contrast medium administration. All patients were instructed not to breathe during the CT scans to prevent motion artefacts. The technical parameters used are reported in Table 1.

In one patient, an acquisition with the axial technique was also performed: thin layer thickness (<1 mm), narrow FOV (13 cm), fine focus, and the double reconstruction filter (bone plus and standard type).

### 2.2. Radiologic Evaluation

For all patients, the following radiological characteristics were analyzed: Hilar–mediastinal lymphadenopathy (short axis greater than 10 mm);Hilar–mediastinal calcified lymph nodes (yes/no);Site of lymphadenopathy (unilateral or bilateral mediastinum, unilateral or bilateral hilum, unilateral or bilateral hilum and mediastinum, in accordance with lung lymphatic drainage [12]);Fissure involvements, defined as the focal or extensive thickening of the interlobar pleura or presence of enlarged lymph nodes within the fissure;Uni- or bilateral pleural effusion;Visceral and parietal pleura involvement, defined as focal or diffuse linear or pseudonodular thickening [13]; classified as not involved, mild, moderate, or severe if it affected, respectively, 0–15%, 15–50%, 50–75%, or >75% of the pleural surface;Pulmonary emphysema;Number of nodules;Predominant location of nodules (right upper lobe, middle lobe, right lower lobe, left upper lobe, left lower lobe);Calcifications within the nodules;Nodule spiculated appearance;Nodule maximum diameter (millimeters), defined on the anatomical plane that offered optimal representation (native axial and 2D multiplanar reconstructed coronal or sagittal planes);Distance of the suspicious lung nodule from the interlobar fissure (millimeters);Lung nodule density (Hounsfield Units, HU).

This process was conducted using a dedicated workstation for multiplanar reconstruction, performed by a single radiologist with more than 10 years of experience in thoracic imaging. During the evaluation of the radiological characteristics of each patient’s nodule, the radiologist was not aware of the definitive diagnosis of the nodule under analysis. Nodule assessment used a fixed value for the lung window, with a width of 1400 HU and a level of −600 HU. The nodule density was assessed on the noncontrast scans, which were available for all 23 patients. For density analysis, a region of interest (ROI) was placed on the nodule comprising as large and homogeneous an area as possible. An attempt was made to place ROIs over the entire nodule, excluding peripheral areas and surrounding structures. Then, for each ROI, the average attenuation value (HU) was recorded. A valid density measurement was ensured by the use of a small FOV (9 cm), thus increasing the spatial resolution.

The 23 patients were then divided into two groups based on histological diagnosis: Group A, confirmed lung neoplasia, and Group B, diagnosed with benign sarcoid nodules. The two study groups were compared based on the previously listed radiological characteristics.

### 2.3. Statistical Analysis

Parametric data are presented as mean ± standard deviation (SD), and nonparametric data as medians with interquartile ranges (IQR). Categorical variables are presented as either a percentage of the total or numerically, as appropriate. Multiple comparisons were assessed by nonparametric one-way ANOVA (Kruskal–Wallis test) and the Dunn test, while the Mann–Whitney U test was performed to compare continuous variables between the two groups. Chi-squared or Fischer’s exact tests were assessed for categorical variables. The level of statistical significance was set at a *p*-level ≤ 0.05. Statistical analyses were performed using the Jamovi 2.3.1 software.

## 3. Results

### 3.1. Patients’ Characteristics

Between January 2010 and January 2023, 23 patients (mean age 54.5 years old, range 32–82 years; 16 men, 69.6%, and 7 women, 30.4%) with sarcoidosis, undergoing regular follow-up, were included in our retrospective single-center study for the presence of single or multiple lung nodules suspicious for LC. The patient’s characteristics are summarized in Table 2.

Among these patients, 4 were smokers (17.4%), 1 a nonsmoker (4.3%), and 18 former smokers (78.3%), defined as those patients with a confirmed smoking history, with the cessation of the smoking habit for at least 6 months at the time of the first radiological investigation performed for follow-up entry. Patients were divided into two groups based on histopathological diagnosis: Group A (*n* = 6), patients with neoplastic nodules, and Group B (*n* = 17), patients with benign nodules. The clinical and radiological characteristics of the two groups were compared using chest CT scans conducted during the follow-up period.

### 3.2. Comparison Between Group A and B

Regarding the clinical characteristics, a statistically significant difference emerged in terms of age, with patients in Group A being older (mean ages 65.3 and 50.7 years old for Groups A and B, respectively, *p* = 0.006). There were no differences between the two groups in terms of sex, smoking habits, or Scadding’s sarcoidosis staging [11]. This allows us to exclude any bias in the smoke-related pathogenesis of LC in the study group and to obtain two homogeneous groups in terms of sarcoidosis interstitial involvement. Table 3 describes the clinical characteristics of the patients in the study for comparison. 

Table 4 describes the surgical and histopathological characteristics of patients in Group A. All patients underwent surgical biopsy with intraoperative frozen section diagnosis and subsequent lung resection.

In four patients of Group A, the analysis of HRCT and 18F-FDG-PET showed positivity of the hilar lymph nodes (cN1), which later resulted in negative and, thus, free-of-tumor cells in the definitive histological examination. Regarding the radiological characteristics of the HRCT, continuous variables are summarized in Table 5 and Table 6. 

A statistically significant difference was observed in lung nodule density, with median values of 60.0 HU for Group A and −126.75 HU for Group B (*p* < 0.0001). Additionally, a significant difference was found in the median diameter of the lung nodules: 23.5 mm in Group A and 12.2 mm in Group B (95% CI 1.359–21.288, *p* = 0.028). Lastly, no significant difference was observed in the distance from the nearest interlobar fissure (median 40.0 mm and 16.3 mm for Group A and B, respectively, 95% CI −4.8118–55.1243, *p* = 0.095); although, the obtained value is close to significance.

Nodule spiculation was found to be a statistically significant variable, with neoplastic nodules more commonly of a spiculated shape compared to sarcoid nodules (spiculated nodules in 100% and 37.5% of patients in Group A and B, respectively, *p* < 0.05). It was not possible to perform a statistical analysis of the differences in the spiculated appearance of the nodules, particularly between the smooth and micronodular variants, the latter of which may be more frequently associated with sarcoidosis, possibly due to the presence of multiple satellite microgranulomas surrounding a primary central lesion. This limitation was due to the lack of sufficient high-resolution spatial imaging. Specifically, we did not have access to a sufficient number of images acquired with the optimal CT parameters, such as minimum slice thickness and, particularly, a narrow FOV acquisition or reconstruction, necessary to accurately assess these fine radiological features.

The number of nodules present on the HRCT was also found to be significant: a single parenchymal nodule was more frequently observed in the neoplastic patient group (*p* = 0.043). No significant difference was observed between the two groups in terms of the presence of lymphadenopathy nor in the localization of enlarged lymph nodes; in particular, in Group A, patients’ hilar and mediastinal lymphadenopathy was widespread, not only affecting the lymph node drainage stations of the LC-affected lobe. No statistically significant differences were found between the other analyzed variables; although, the presence of pulmonary emphysema appeared to be significant (*p* = 0.0056), with a higher representation of pulmonary emphysema in patients with LC. The analyzed dichotomous radiological variables are summarized in Table 5.

In Figure 1a–d and Figure 2a–e, we show some images from our patients.

To differentiate patients requiring surgery from those suitable for close follow-up of suspicious lung nodules—defined as nodules that persist or grow during radiological follow-up—we developed a diagnostic algorithm. The algorithm’s cut-offs are based on statistically significant differences in variables between the two study groups. It recommends surgical referral under the following conditions:Spiculated nodules, regardless of size, density, or number;Nonspiculated nodules larger than 20 mm or with a density ≥ 60 HU, whether presenting as a single or multiple parenchymal lesion;Nonspiculated nodules measuring 15–20 mm and/or with a density between 45 and 60 HU but presenting as a single lung lesion.

Since no significant difference was observed in terms of lymph node enlargement, lymph node calcification, or lymph node location between the two groups, these were not taken into consideration for the compilation of the algorithm. The proposed algorithm is summarized in Figure 3.

## 4. Discussion

Patients with sarcoidosis appear to have an increased risk of developing cancer, particularly in the organs affected by the disease [6]. However, survival at 36 months has been shown to be similar in patients with sarcoid interstitial lung disease with and without lung cancer (LC) [14]. Several cases of coexisting LC and sarcoidosis have been described [15,16,17,18], though the mechanism linking these two conditions remains unclear [19]. In our study group, LC developed in patients who were diagnosed with sarcoidosis an average of 55 months earlier, consistent with findings reported in the literature. However, in some cases [20,21], the diagnosis of sarcoidosis and LC is simultaneous, not allowing us to determine whether sarcoidosis is the cause of tumor development from chronic inflammatory stimulation or if LC has led to the subsequent development of sarcoidosis by inducing a tumor-mediated immune histological response.

Pulmonary sarcoidosis can exhibit distinctive radiological characteristics, including isolated parenchymal lesions, which can complicate the differential diagnosis with lung cancer (LC). These solitary lesions can pose a diagnostic challenge, requiring careful evaluation to distinguish pulmonary sarcoidosis from LC. Differential diagnosis between sarcoid and cancer nodules is especially challenging considering that both conditions often share hilar–mediastinal lymph node involvement. Histological diagnosis in sarcoidosis is often challenging because the lung parenchyma, which is chronically exposed to inflammatory insults, is susceptible to complications, such as pneumothorax and hemothorax during transcutaneous or surgical biopsies, and there is also a risk of false-negative results [20]. Our study aims to identify radiological features that can aid in the differential diagnosis between sarcoid-like nodules and lung cancer. This information is crucial for guiding clinical decisions, helping to identify patients with a high suspicion of malignancy who may benefit from surgical resection, while avoiding unnecessary surgery in those with benign disease. To the best of our knowledge, this is the first study to compare the HRCT characteristics of patients with nodular sarcoidosis and concomitant lesions suspicious for LC. With a total of six patients, our single-center study has the largest number of subjects with a diagnosis of LC and a previous diagnosis of sarcoidosis, as most studies conducted so far have been case reports of individual cases [9,22,23,24]. To date, the study with the most reported cases has been that of Yamasawa et al. [25], in which, however, the authors do not describe in detail the radiological characteristics observed using HRCT in the patients included (*n* = 4). In our work, patients diagnosed with LC were older compared to the control group. This is consistent with the natural history of LC development based on chronic inflammatory stimulation [26]. Patients diagnosed with sarcoidosis are young, often being diagnosed at ages between 20 and 40. It is plausible that the presence of prolonged inflammatory stimulation over the years could allow for the onset of LC, which, however, occurs at a much later age. 

Regarding the radiological characteristics during HRCT, nodules in the LC Group A had a larger diameter compared to Group B, with a statistically significant difference. Similarly, neoplastic nodules in Group A had a higher density and more frequently appeared as single parenchymal nodules. Finally, it was observed that all patients in Group A had a nodule with a spiculated appearance.

The results obtained are related to the accurate characterization of nodular lesions through the use of an examination technique with high spatial resolution (for the study of morphology and margins) and high contrast resolution (for the study of intranodular densities). Indeed, this type of study has made it possible to obtain images of high diagnostic quality that facilitate the differential diagnosis of pulmonary nodules in most cases [27]. This study allows for the detection and characterization of ground/glass and especially mixed nodules (solid and ground/glass) for which even the Fleischner Society has developed guidelines for the follow-up of these lesions based on size and composition [28,29,30]. Although several of the analyzed characteristics are already known indicators of malignancy in parenchymal nodules in patients without sarcoidosis (e.g., nodule spiculation and size), the density of the nodule measured with the baseline CT was found to be the most discriminative parameter between sarcoidotic and malignant nodules. This factor, which is currently considered secondary, emerged as fundamentally important in our study.

The validity of our algorithm is supported by the fact that several studies [25,31] have already demonstrated that LCs arising in sarcoidosis-affected parenchyma tend to grow slowly, without generating lymph node metastases. Our study also showed the absence of lymph node metastases in the four cases where lymph node involvement was suspected at the time of radiological diagnosis (cN1). Furthermore, in Patients 3 and 5, the stability of the lung nodule was observed for almost 10 years, indicating very slow growth. In Patient 1, 4 years from the initial diagnosis, the LC showed an increase in size, without any lymph node involvement, as confirmed by histological examination. 

A limitation of this study is the small sample size, necessitating further research with larger cohorts to reinforce the conclusions drawn. The small number of nodules studied after the administration of contrast medium did not allow for a statistically valid assessment of nodular characteristics after contrast medium. Additionally, the recurrence rate of lung cancer within patients was not analyzed; future studies should also assess disease-free survival and overall survival in these patients. 

## 5. Conclusions

An accurate clinical evaluation and HRCT are essential for diagnosing lung cancer in sarcoidotic patients and determining candidacy for surgical intervention. Factors such as older age, spiculated nodule morphology, higher nodule density on the HRCT, larger nodule size, and the presence of a single nodule rather than multiple nodules appear to be associated with a malignant nature. Nevertheless, sarcoidosis seems to slow the growth of LC and lymph node spread. Radiological follow-up could be a valid initial option to surgery in patients with several comorbidities and without a definite nodule diagnosis, without compromising the following surgical radicality.

## Figures and Tables

**Figure 1 diagnostics-14-02389-f001:**
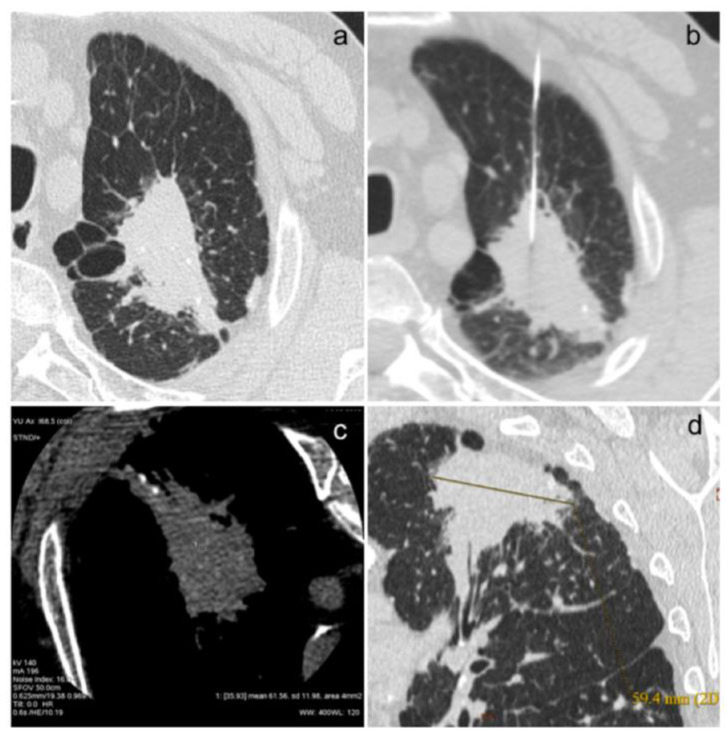
(**a**–**d**). This is the case of an 80-year-old patient with acinar (40%) and solid (20%) adenocarcinoma with in situ areas of lepidic morphology (40%) and adenocarcinomatous infiltration of the visceral pleura (**a**), confirmed on biopsy examination (**b**). The neoplastic lesion has a high density (61.56 HU, (**c**)) and a size of about 60 mm (**d**).

**Figure 2 diagnostics-14-02389-f002:**
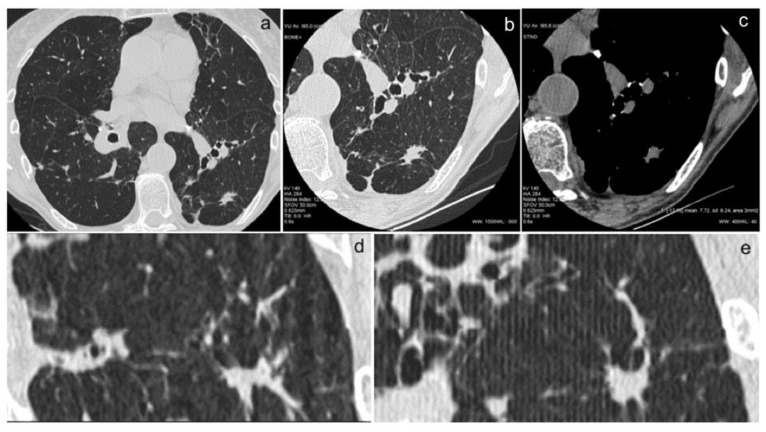
(**a**–**e**). The use of an examination technique with bone plus kernel reconstruction (for the study of lung parenchyma) (**a**), high spatial resolution (for the study of morphology and margins) (**b**), and high contrast resolution (for the study of intranodular densities) (**c**) allow for an accurate characterization of nodular lesions. This type of study makes it possible to obtain images of high diagnostic quality also in coronal (**d**) and sagittal reconstructions (**e**), enabling the differential diagnosis of pulmonary nodules.

**Figure 3 diagnostics-14-02389-f003:**
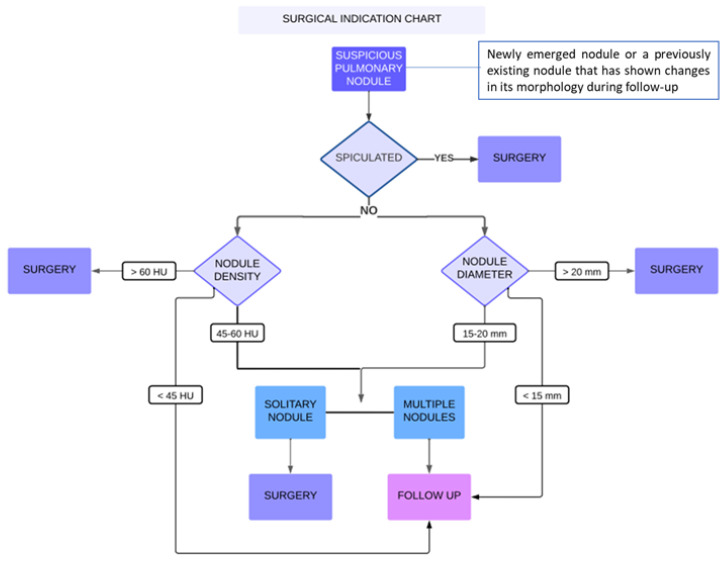
Decision algorithm on surgical indication in patients with sarcoidosis and suspicious lung nodule (defined as new-onset nodules that persist or grow during radiological follow-up regardless medical therapy) in the absence of a diagnosis.

**Table 1 diagnostics-14-02389-t001:** Technical parameters of CT scans.

Parameters	HRCT	Postcontrastographic Chest CT Scan
Slide thickness (mm)	3.75	1.25
Reconstruction interval (mm)	1.25	0.6
mAs	250/400	250/640
kVp	140	140
Revolution time (s)	0.6	0.6
Index noise	16	16
FOV	Encompassed on both lungs	Encompassed on both lungs
Reconstruction kernel	High spatial resolution (bone plus) for parenchyma and standard algorithm for mediastinal evaluation	standard
Collimation (mm)	20 (small focal spot)	40
Postcontrast phase	-	Late arterial (45–50 s)

CT: computed tomography; FOV: field of view, HRCT: high-resolution computed tomography.

**Table 2 diagnostics-14-02389-t002:** Patients’ characteristics.

Variable ^a^	Number (Range)	%
Sex		
F	7	30.4
M	16	69.6
Age	54.5 (32–82)	
Smoking status		
Smoker	4	17.4
Nonsmoker	1	4.3
Former smoker	18	78.3
Scadding Stage		
0	5	21.7
1	3	13
2	7	30.4
3	7	30.4
4	1	4.3
Biopsy approach		
FNAB	12	52.1
VATS	6	26.1
Thoracotomy	3	13.1
RATS	2	8.7

^a^ Data are shown as a percentage for categorical variables and as medians for continuous variables. FNAB: fine-needle aspiration biopsy; RATS: robotic-assisted thoracoscopy; VATS: video-assisted thoracoscopy.

**Table 3 diagnostics-14-02389-t003:** Patients’ characteristics and radiological features for study comparison (Group A and Group B).

Variable ^a^	Group A (*n* = 6)	Group B (*n* = 17)	*p*-Value
Sex			0.858
F	2 (33.3%)	5 (29.4%)	
M	4 (66.7%)	12 (70.6%)	
Age	65.3 (IQR 11.5)	50.7 (IQR 18.00)	**0.006**
Smoking status			0.432
Smoker	0 (0.0%)	1 (5.9%)	
Nonsmoker	2 (33.3%)	2 (11.8%)	
Former smoker	4 (66.7%)	14 (82.4%)	
Scadding Stage			0.206
0	2 (33.3%)	3 (17.6%)	
1	2 (33.3%)	1 (5.9%)	
2	0 (0.0%)	7 (41.2%)	
3	2 (33.3%)	5 (29.4%)	
4	0 (0.0%)	1 (5.9%)	

^a^ Data are shown as a percentage for categorical variables and medians with interquartile range (IQR) for continuous variables. The significant variables are shown in bold.

**Table 4 diagnostics-14-02389-t004:** Group A surgical and histopathological characteristics.

Patients	Resection	Approach	Surgery Time (min)	Intraoperative Complications	Histology	pT	pN
P1	LUL lobectomy	Open	210	Yes	ADK	T2b	0
P2	RML lobectomy	VATS	220	Yes	Clear cell carcinoma	/	0
P3	LLL lobectomy	VATS	135	No	ADK	T1b	0
P4	LLL atypical resection	VATS	50	No	ADK	T3	0
P5	LUL typical resection	RATS	175	No	ADK	T1m	0
P6	RUL + RLL atypical resections	RATS	65	No	Carcinoid	T1	0

ADK: adenocarcinoma; Min: minutes; P: patient; RUL: right upper lobe; RML: right middle lobe; RLL: right lower lobe; LUL: left upper lobe; LLL: left lower lobe; RATS: robotic-assisted thoracoscopic surgery; VATS: video-assisted thoracoscopic surgery.

**Table 5 diagnostics-14-02389-t005:** Nodules’ radiological appearance (categorical variables).

Variable ^a^	Group A (*n* = 6)	Group B (*n* = 17)	*p*-Value
Spiculated nodule	6 (100%)	6 (37.5%)	**0.012**
Emphysema	4 (66.7%)	4 (23.5%)	0.06
Nodule calcification	1 (16.7%)	8 (47.1%)	0.19
Nodule location			0.18
RUL	1 (16.7%)	6 (37.5%)	
RML	1 (16.7%)	0 (0.0%)	
RLL	0 (0.0%)	4 (23.5%)	
LUL	2 (33.3%)	3 (17.6%)	
LLL	2 (33.3%)	1 (5.9%)	
RUL + RLL	0 (0.0%)	1 (5.9%)	
RUL + LUL	0 (0.0%)	2 (11.8%)	
Number of nodules			**0.043**
Single	5 (83.3%)	6 (35.3%)	
Multiple	1 (16.7%)	11 (64.7%)	
Pleural thickening			0.17
Absent	2 (33.3%)	4 (23.5%)	
Mild	1 (16.7%)	10 (58.8%)	
Moderate	1 (16.7%)	0 (0.0%)	
Severe	2 (33.3%)	3 (17.6%)	
Pleural effusion	1 (16.7%)	2 (11.8%)	0.76
Fissure involvement	3 (50%)	4 (23.5%)	0.23
Lymph node enlargement	4 (66.7%)	11 (64.7%)	0.93
Lymph node location			0.09
No lymph nodes	1 (16.7%)	4 (23.5%)	
Ilum, bilateral	0 (0.0%)	1 (5.9%)	
Mediastinum, bilateral	2 (33.3%)	3 (17.6%)	
Ilar, bilateral + med.	1 (16.7%)	9 (52.9%)	
Ilar, monolateral, med.	2 (33.3%)	0 (0.0%)	
Lymph node calcification	2 (33.3%)	8 (47.1%)	0.56

^a^ Data are shown as a percentage for categorical variables. RUL: right upper lobe; RML: right middle lobe; RLL: right lower lobe; LUL: left upper lobe; LLL: left lower lobe; med: mediastinum. The significant variables are shown in bold.

**Table 6 diagnostics-14-02389-t006:** Nodules’ radiological appearance (continuous variables)

Variable ^a^	Group A (*n* = 6)	Group B (*n* = 17)	Lower CI	Upper CI	*p*-Value
Nodule diameter (mm)	23.5 (IQR10.5)	12.2 (IQR7.0)	1.359	21.288	**0.028**
Fissure distance (mm)	40.0 (IQR29.5)	16.3 (IQR26.0)	−4.812	55.124	0.095
Nodule density (HU)	60.02 (IQR32.05)	−126.75 (IQR158.63)	70.130	303.421	**<0.001**

^a^ The data are shown as a percentage for categorical variables and as medians (IQR) for continuous variables. The significant variables are shown in bold.

## Data Availability

The original contributions presented in the study are included in the article; further inquiries can be directed to the corresponding author.

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
