# Peer review of "Sarcoid Nodule or Lung Cancer? A High-Resolution Computed Tomography-Based Retrospective Study of Pulmonary Nodules in Patients with Sarcoidosis"

_diagnostics, 2024, doi:10.3390/diagnostics14212389_

Round 1
Reviewer 1 Report
Comments and Suggestions for Authors
Thank you for the opportunity to evaluate the article “Sarcoid nodule or lung cancer? A high-resolution computed tomography-based retrospective study of pulmonary nodules in sarcoidosis patients”. The authors present an interesting article comparing the characteristics of sarcoid and neoplastic nodules on high-resolution non-enhanced computed tomography, also suggesting a decision algorithm.
There are several aspects that must be addressed, especially in the Materials and Methods section, which may impact the credibility of the data presented in the Results section and invalidate the decision algorithm if not explained properly.
1. The technical parameters for the CT scan should also include the pitch value used for the acquisition.
2. Was there any standardized quality control method/algorithm used to evaluate if the CT scan images are suitable for interpretation? If yes please describe it in Materials and Methods. If not, please mention this aspect in the study limitation subsection.
3. The density measurement modality for the lung nodules is not described in the Materials and Methods section.
a) Did you use a fixed-dimension circular ROI for all nodules? or b) Did you manually trace the limit of the nodules and measure the density considering the whole section of the nodule? or c) Did you perform a volumetric measurement for the whole nodule, including dimensions and density?
4. Did you account for the partial volume effect from the surrounding lung during density measurement? This effect can falsely lower the density of a lung nodule, especially if the nodule is small. Please specify in the Materials and Methods section how you ensured a valid, unbiased, and reproducible density measurement for all nodules, regardless of their dimensions.
5. Sarcoidosis is often associated with various patterns of lung fibrosis and consolidation areas. However, these changes were not included in the radiologic evaluation parameters mentioned in subsection 2.3 of the Materials and Methods, although, fibrotic changes are visible in the images included in the article. The characteristics mentioned above should be introduced for the 23 patients included in the study and added to the radiological findings described in Table 5."
6. Iodinated contrast administration could improve imaging diagnostic accuracy, reducing the need for biopsy or follow-up examination to determine the growth rate of the nodule. However, all the radiological characteristics described in the Results section are on the non-contrast scans. This aspect also should be mentioned in the study limitation subsection.
Author Response
Response to Reviewer #1
Thank you for the opportunity to evaluate the article “Sarcoid nodule or lung cancer? A high-resolution computed tomography-based retrospective study of pulmonary nodules in sarcoidosis patients”. The authors present an interesting article comparing the characteristics of sarcoid and neoplastic nodules on high-resolution non-enhanced computed tomography, also suggesting a decision algorithm.
There are several aspects that must be addressed, especially in the Materials and Methods section, which may impact the credibility of the data presented in the Results section and invalidate the decision algorithm if not explained properly.
- The technical parameters for the CT scan should also include the pitch value used for the acquisition.
We sincerely thank the reviewer for the kind consideration in our manuscript and the interesting comments. The pitch value is reported in the text as ‘revolution time’. (Table 1, line 5).
- Was there any standardized quality control method/algorithm used to evaluate if the CT scan images are suitable for interpretation? If yes please describe it in Materials and Methods. If not, please mention this aspect in the study limitation subsection.
Thank you for your comment, the evaluability of the images was defined by the radiologist for each CT scan, excluding a priori those patients for whom the imaging was not suitable for evaluation.
This ‘exclusion criterion’ was included in materials and methods, highlighted in yellow. (Page 2, line 86-88).
- The density measurement modality for the lung nodules is not described in the Materials and Methods section.
- a) Did you use a fixed-dimension circular ROI for all nodules? or b) Did you manually trace the limit of the nodules and measure the density considering the whole section of the nodule? or c) Did you perform a volumetric measurement for the whole nodule, including dimensions and density?
Thank you for your comment, the missing parts (density measurement modality) have been added to the text in the ‘materials and methods’ section, highlighted in yellow. (Page 3, lines 145-149).
- Did you account for the partial volume effect from the surrounding lung during density measurement? This effect can falsely lower the density of a lung nodule, especially if the nodule is small. Please specify in the Materials and Methods section how you ensured a valid, unbiased, and reproducible density measurement for all nodules, regardless of their dimensions.
Thank you for your comment, A valid, unbiased and reproducible density measurement for all nodules, irrespective of their size, has been achieved by using a small FOV, thereby increasing spatial resolution, this have been added to the text in the ‘materials and methods’ section, highlighted in yellow. (Page 3, lines 144-145).
- Sarcoidosis is often associated with various patterns of pulmonary fibrosis and areas of consolidation. However, these changes were not included in the radiological evaluation parameters mentioned in subsection 2.3 of Materials and Methods, although fibrotic changes are visible in the images included in the article. The above-mentioned features should be introduced for the 23 patients included in the study and added to the radiological findings described in Table 5’.
Thank you for your comment. The fibrotic changes associated with sarcoidosis were included in the analysis in the form of the ‘scadding stage’ analysis, shown in the results in table 2 (page 5)
- Iodinated contrast administration could improve imaging diagnostic accuracy, reducing the need for biopsy or follow-up examination to determine the growth rate of the nodule. However, all the radiological characteristics described in the Results section are on the non-contrast scans. This aspect also should be mentioned in the study limitation subsection.
Thank you for your comment. The aim of this study is to compare the characteristics of sarcoid nodules and neoplastic nodules at HRCT, for this reason and because the patients with CT before and after mdc are only 7/23, the post-contrastographic characteristics of the nodules were not evaluated. The small sample size would have made the statistics unreliable anyway. However, this consideration has been added within the limits, highlighted in yellow. (page 11, lines 327-329).
Reviewer 2 Report
Comments and Suggestions for Authors
My comments:
A very small group of patients was studied. This was included in the LIMITATIONS chapter, but concluding on this basis is difficult
There is no information about the characteristics of neoplastic and non-neoplastic nodules in the first CT scans
There is no information about the doubling growing time, volumetric assessment of nodules
There is no statistical information about the change in nodules in the entire evaluated population (23 patients)
Perhaps the authors could try to collect data from other centers and create 3-4 center retrospective studies
It is worth including a paragraph about the possibility of using photon counting tomography in the analysis.
Author Response
A very small group of patients was studied. This was included in the LIMITATIONS chapter, but concluding on this basis is difficult.
Thank you for your comment. We understand the smallness of the sample, but there are no similar case studies on the subject in the literature, so we felt it was important to report our results.
There is no information about the characteristics of neoplastic and non-neoplastic nodules in the first CT scans. There is no information about the doubling growing time, volumetric assessment of nodules. There is no statistical information about the change in nodules in the entire evaluated population (23 patients)
Thank you for your comments. The characteristics of the nodules were analysed on the last HRCT before surgery, to take a ‘snapshot’ of the nodule features that were when most correlated with the diagnosis of certainty. For this reason, there are no information about change in nodules. An analysis on volumetric doubling was not performed, but an evaluation on maximum nodular diameters, as reported in “Materials and Methods”.
Perhaps the authors could try to collect data from other centers and create 3-4 center retrospective studies.
Thank you for your comments. This is a very good idea and it would be our intention to involve other sarcoidosis referral centres to increase the caseload.
It is worth including a paragraph about the possibility of using photon counting tomography in the analysis.
Thank you for your comments. We do not have a photon-counting scan in our centre; in our opinion, to put in a sentence just to say that it would be useful would not be clinically relevant.
Reviewer 3 Report
Comments and Suggestions for Authors
Dear Editor,
The authors compared the characteristics of sarcoid nodules and neoplastic nodules on high-resolution computed tomography (HRCT) in sarcoidosis patients. Between 2010 and 2023, among 685 patients affected by pulmonary sarcoidosis, twenty-three patients developed pulmonary nodules of suspected malignant nature. The results of the study emphasized that an accurate clinical evaluation and HRCT examination are crucial to diagnose lung cancer in patients with sarcoidosis and to determine who requires surgical resection. Congratulations to the authors for this beautiful work. It will be read with interest and is an important article for follow-up in sarcoidosis patients.
Sincerley
Comments on the Quality of English LanguageModerate editing of English language required.
Author Response
Comment 1:
Dear Editor,
The authors compared the characteristics of sarcoid nodules and neoplastic nodules on high-resolution computed tomography (HRCT) in sarcoidosis patients. Between 2010 and 2023, among 685 patients affected by pulmonary sarcoidosis, twenty-three patients developed pulmonary nodules of suspected malignant nature. The results of the study emphasized that an accurate clinical evaluation and HRCT examination are crucial to diagnose lung cancer in patients with sarcoidosis and to determine who requires surgical resection. Congratulations to the authors for this beautiful work. It will be read with interest and is an important article for follow-up in sarcoidosis patients.
Sincerley.
Answer 1: Dear Editor,
Thank you for your kind words and for recognizing the importance of our study. We are grateful for your appreciation of the clinical relevance of differentiating between sarcoid and neoplastic nodules using high-resolution computed tomography (HRCT) in sarcoidosis patients. As you pointed out, this topic remains underexplored in the current literature, and we hope that our findings will contribute to raising awareness and advancing knowledge in this area. Your thoughtful feedback encourages us to continue investigating this crucial yet often overlooked aspect of sarcoidosis management.
Comment 2: Moderate editing of English language required.
Thank you for your comments. English has been further revised.
Round 2
Reviewer 1 Report
Comments and Suggestions for Authors
Thank you for the opportunity to evaluate the article “Sarcoid nodule or lung cancer? A high-resolution computed tomography-based retrospective study of pulmonary nodules in sarcoidosis patients”.
The authors addressed all the concerns expressed in the previous review, and I recommend the acceptance for publication of the article.
Reviewer 2 Report
Comments and Suggestions for Authors
I have no further comments.